# OpenReview forum: "Cross-Modal Obfuscation for Jailbreak Attacks on Large Vision-Language Models"
_ICLR.cc/2026/Conference — ICLR 2026 Conference Withdrawn Submission_

### Official Review · Reviewer_HsAs · 2025-10-16

**Soundness:** 2
**Presentation:** 2
**Contribution:** 2
**Rating:** 2
**Confidence:** 4

**Summary:**

The paper proposed Cross-modal Adversarial Multimodal Obfuscation (CAMO), a black-box jailbreak method on VLMs, which used cross-modal reasoning to reconstruct attack intent.

**Strengths:**

1. The paper is easy to follow.
2. The method is a black-box attack and it is easy to implement.

**Weaknesses:**

1. This method applied a widely-used jailbreak strategy (reasoning distraction) in the cross-modal setting, which lacks novelty.
2. The paper claimed that "CAMO decomposed harmful instructions into benign-looking textual and visual clues", but from Figure 2, it seems that the visual input is still harmful (a bomb).
3. This paper is only evaluated on AdvBench family datasets, which doesn't effectively prove the validity of CAMO.
4. The baseline methods compared in the paper are not the current strong jailbreak methods. The authors should include comparisons with the latest strong methods (AutoDAN [1], UMK [2], and BAP Attack [3], etc.) to validate the effectiveness of CAMO.
5. This method relies on a sensitive dictionary to select keywords, so if a harmful input is not included in this dictionary, the method may fail. Therefore, it seems to lack transferability.
6. The paper evaluates the method only on models with over 70B parameters and does not test the attack on models with 7B, 8B, or 13B parameters. Since models with 7B/13B parameters seem unlikely to follow the Adversarial Prompt Template, the authors should include attack results on 7B, 8B and 13B models to demonstrate the effectiveness of the attack method on smaller models.
7. The authors state in the abstract and introduction that CAMO achieves an attack success rate of 81.82% on GPT-4.1-nano and 93.94% on DeepSeek-R1. However, these are only ASR values for a single category from one dataset, and do not reflect the overall ASR. The authors should revise this and clarify it.
8. In the 'Attack Against Defense' section, the perplexity filter, OCR, and OpenAI's moderation are not VLM jailbreak defense methods, and OpenAI's moderation has a very broad definition for harmful content. Therefore, these methods do not effectively test CAMO's robustness. The authors should include strong jailbreak defense methods for VLMs, such as Adashield [4] and BlueSuffix [5].
9. The current evaluation only relies on ASR. The authors should include additional metrics, such as using the Perspective API, to provide a more comprehensive assessment.

[1] AutoDAN: Generating Stealthy Jailbreak Prompts on Aligned Large Language Models. ICLR 2024.

[2] White-box multimodal jailbreaks against large vision-language models. ACM MM, 2024.

[3] Jailbreak Vision Language Models via Bi-Modal Adversarial Prompt. IEEE TIFS, 2025.

[4] AdaShield: Safeguarding Multimodal Large Language Models from Structure-based Attack via Adaptive Shield Prompting. ECCV, 2024.

[5] BlueSuffix: Reinforced Blue Teaming for Vision-Language Models Against Jailbreak Attacks. ICLR, 2025.

**Questions:**

See Weaknesses.

---

### Official Review · Reviewer_f6vV · 2025-10-24

**Soundness:** 2
**Presentation:** 3
**Contribution:** 2
**Rating:** 2
**Confidence:** 4

**Summary:**

This paper proposes CAMO (Cross-modal Adversarial Multimodal Obfuscation), a new black-box jailbreak framework targeting Large Vision-Language Models (LVLMs). The main idea is to decompose harmful instructions into benign textual and visual components—for example, splitting a sensitive word like “bomb” into masked textual fragments and numeric/visual clues embedded in an accompanying image. Although each modality appears harmless in isolation, LVLMs can semantically reconstruct the hidden intent through cross-modal reasoning, effectively bypassing standard input-level safety filters.

**Strengths:**

- The paper proposed a simple but effective jailbreak attack, by nesting the math problem with OCR for smuggling the harmful contents within the MLLM reasoning process.
- The paper is well-written and easy to follow.
- Extensive experiments over both open- / closed-source models with ablation studies and analysis, confirming the effectiveness of the proposed jailbreak attack.

**Weaknesses:**

- The paper’s core idea of obfuscating harmful text and guiding the model to reconstruct it is conceptually similar to prior jailbreak and text-obfuscation methods such as [R1]. The overall attack logic—hiding malicious intent through structured obfuscation and letting the model decode it internally—is not substantially new. The novelty thus feels incremental, focusing on implementation details rather than a fundamentally new attack paradigm.
- Despite the paper’s claim that CAMO significantly reduces computational cost, it is unclear how such reduction is achieved, since the method does not seem to introduce any specific mechanism or contribution designed to lower computation. The reported token savings mainly come from comparing against baselines that require additional optimization or multi-turn generation steps. When compared to similar baselines that use lightweight single-query methods (e.g., FigStep), the efficiency advantage might be marginal.
- The obfuscation appears to occur on the textual side, while the visual component mainly serves as an auxiliary lookup table. In practice, a safety filter could still recognize harmful semantics from the image itself. This raises questions about whether the approach truly provides multimodal stealth.
- For the text-only method, masking long or compound attack target phrases (e.g., “bomb or explosives for killing someone”) might render the reconstruction impossible or misinterpreted. How can such lengthy-word attack queries be ensured to be reconstructed by MLLMs?
- The paper assumes that once the model reconstructs the hidden harmful word, it will continue generating the harmful response. I wonder if it would also be effective against the output-side moderation filters, which might easily detect the reconstructed harmful words and reject the attack query.
- In Table 1, why the proposed text-only attack shows significantly lower performance than the baseline (-20%) in KS scenario?

[R1] Playing the Fool: Jailbreaking LLMs and Multimodal LLMs with Out-of-Distribution Strategy

**Questions:**

See above weaknesses.

---

### Official Review · Reviewer_rGer · 2025-10-25

**Soundness:** 3
**Presentation:** 3
**Contribution:** 3
**Rating:** 4
**Confidence:** 4

**Summary:**

The paper proposes to obfuscate sensitive words in harmful prompts using a combination of character masking with easy math problem solving. Using this technique, a harmful prompt is split into benign text and benign visual prompts, which is not detectable by basic techniques like perplexity filtering for text and OCR detection for images. The authors show high ASR for closed sourced and open sourced models as measured by GPT4o as judge.

**Strengths:**

As mentioned in the summary, the authors come up with a novel attack strategy that is good at obfuscation of sensitive words in harmful prompts and are able to demonstrate very high ASR across several models, which is a great experimental result. I am impressed by the strong empirical numbers.

**Weaknesses:**

Despite the authors checking the robustness of their attack against 3 filters: perplexity, OCR, and OpenAI moderation tool -- I am interested in seeing the robustness against open-sourced guard models, like Llamaguard, WildGuard, AegisGuard (D and P). And I know the input is multimodal whereas these guard models are unimodal, so can the authors just check this output of these guard models for the text part of the input and report those experimental results.

I am also interested in seeing the results against multimodal guard models like: https://docs.nvidia.com/nemo/guardrails/latest/user-guides/multimodal.html, https://huggingface.co/meta-llama/Llama-Guard-4-12B and reasoning based guard models like https://huggingface.co/yueliu1999/GuardReasoner-8B or others if authors can find them. It is important to show that the attack is robust against both existing text based, multimodal, and reasoning based guard models.

**Questions:**

I have asked the authors to perform additional experiments to show the robustness of the attacks in the weaknesses section. Please refer to it.

---

### Note · Authors · 2025-11-27

**Comment:**

Thank you very much for your time in handling our submission and for the helpful feedback provided by the reviewers. After careful consideration, we have decided to withdraw our paper from ICLR at this time. We will carefully incorporate the reviewers’ comments to further improve our method and conduct additional experiments, with the aim of preparing a more complete and rigorous version for future submission.

**Withdrawal Confirmation:**

I have read and agree with the venue's withdrawal policy on behalf of myself and my co-authors.